

# Using high-throughput sequencing of ITS2 to describe *Symbiodinium* metacommunities in St. John, US Virgin Islands

Ross Cunning[1], Ruth D. Gates[1] and Peter J. Edmunds[2]

[1] Hawai'i Institute of Marine Biology, University of Hawai'i at Mānoa, Kāne'ohe, HI, United States of America
[2] Department of Biology, California State University, Northridge, CA, United States of America

## ABSTRACT

Symbiotic microalgae (*Symbiodinium* spp.) strongly influence the performance and stress-tolerance of their coral hosts, making the analysis of *Symbiodinium* communities in corals (and metacommunities on reefs) advantageous for many aspects of coral reef research. High-throughput sequencing of ITS2 nrDNA offers unprecedented scale in describing these communities, yet high intragenomic variability at this locus complicates the resolution of biologically meaningful diversity. Here, we demonstrate that generating operational taxonomic units by clustering ITS2 sequences at 97% similarity within, but not across, samples collapses sequence diversity that is more likely to be intragenomic, while preserving diversity that is more likely interspecific. We utilize this 'within-sample clustering' to analyze *Symbiodinium* from ten host taxa on shallow reefs on the north and south shores of St. John, US Virgin Islands. While *Symbiodinium* communities did not differ between shores, metacommunity network analysis of host-symbiont associations revealed *Symbiodinium* lineages occupying 'dominant' and 'background' niches, and coral hosts that are more 'flexible' or 'specific' in their associations with *Symbiodinium*. These methods shed new light on important questions in coral symbiosis ecology, and demonstrate how application-specific bioinformatic pipelines can improve the analysis of metabarcoding data in microbial metacommunity studies.

Corresponding author
Ross Cunning, cunning8@hawaii.edu

## INTRODUCTION

The composition of symbiotic algal communities (*Symbiodinium* spp.) in reef-building scleractinian and milleporine corals plays a major role in their biology and ecology, as identity and the functional performance of symbionts influences emergent properties of the holobiont, including its photobiology, energetics, growth rates, and susceptibility to stress (*Rowan, 2004*; *Cantin et al., 2009*; *Jones & Berkelmans, 2011*). Even slight differences in the relative abundance of different algal symbionts may have important functional consequences for the host (*Loram, Trapido-Rosenthal & Douglas, 2007*; *Cunning, Silverstein & Baker, 2015*; *Bay et al., 2016*). Moreover, variation in these communities among

individuals within a host species, and within individuals over time (*Thornhill et al., 2006*; *Edmunds et al., 2014*), is an important trait modulating sensitivity of corals to environmental stress (*Putnam et al., 2012*), and their ability to respond in beneficial ways to changing environmental conditions by 'shuffling' or 'switching' symbionts (*Baker, 2003*). Therefore, an accurate and comprehensive description of *Symbiodinium* communities within coral hosts informs understanding of the ecological performance of corals.

*Symbiodinium* identity currently is primarily described using genetic sequences of chloroplast (cp23S and psbA$^{ncr}$) and nuclear markers (18S, 28S, ITS1, and ITS2) (*LaJeunesse, 2001*; *Santos, Gutierrez-Rodriguez & Coffroth, 2003*; *Pochon et al., 2012*). Together, these markers have been used to identify nine major 'clades' within the genus *Symbiodinium* (clades A through I (*Pochon & Gates, 2010*)), which have been further divided into 'types' based on the marker with which they are identified. *Symbiodinium* species are beginning to be described based on combinations of markers, including microsatellites to establish reproductive isolation (i.e., to satisfy the biological species concept) (*LaJeunesse et al., 2014*; *Thornhill et al., 2014*). However, ecological surveys of *Symbiodinium* diversity still generally rely on commonly-used marker genes, such as ITS2. Consequently, high-throughput sequencing of ITS2 is being utilized to characterize *Symbiodinium* communities with unprecedented resolution (e.g., *Kenkel et al., 2013*; *Arif et al., 2014*; *Green et al., 2014*; *Quigley et al., 2014*; *Quigley, Willis & Bay, 2016*; *Thomas et al., 2014*; *Edmunds et al., 2014*; *Cunning et al., 2015*; *Ziegler et al., 2017*). However, such analyses often create datasets consisting of millions of sequence reads and hundreds of thousands of distinct sequence variants (*Ziegler et al., 2017*), which places great importance on the ways in which ITS2 sequence diversity is analyzed and interpreted in relation to biological diversity.

While ITS2 initially was investigated as a potential species-level marker in *Symbiodinium* (*LaJeunesse, 2001*), it is now understood that this marker is not sufficiently variable to distinguish all species within this genus (*Finney et al., 2010*; *Thornhill et al., 2014*). For example, the 'B1' ITS2 sequence variant is shared by *S. minutum*, *S. pseudominutum*, and *S. antillogorgium*, and potentially other species (*Parkinson, Coffroth & LaJeunesse, 2015*). Moreover, the position of ITS2 within the tandemly-repeating ribosomal DNA array creates multiple ITS2 sequence variants within a single genome (*Thornhill, LaJeunesse & Santos, 2007*) that evolve through concerted evolution (e.g., *Dover, 1986*). In fact, concerted evolution may mask species divergence within *Symbiodinium* by maintaining ancestral sequence variants as numerical dominants in multiple derived lineages (*Thornhill et al., 2014*). Together, these features of ITS2 complicate the interpretation of intragenomic versus interspecific variation, and preclude its use as a true species-level marker for *Symbiodinium*. Nevertheless, numerically dominant intragenomic variants of ITS2 are still phylogenetically informative across the genus (*LaJeunesse, 2001*), and resolve diversity at a functionally and ecologically important level. Moreover, due to the large quantity of existing sequence data for comparative analysis (e.g., *Franklin et al., 2012*; *Tonk et al., 2013*), and the relative ease of amplifying and sequencing ITS2, it remains an essential and powerful marker for *Symbiodinium*. Therefore, it is important to develop best practices in the bioinformatic analysis and interpretation of ITS2 metabarcoding surveys of *Symbiodinium*.

In addressing this objective, we test the ability of 'within-sample clustering' (i.e., independently clustering sequences at 97% similarity within each sample) to generate biologically relevant operational taxonomic units (OTUs) from ITS2 metabarcoding data. Specifically, this approach addresses the fact that dominant ITS2 sequences from different *Symbiodinium* species may be more similar to one another than intragenomic variants within one *Symbiodinium* (*Thornhill et al., 2014*; *Arif et al., 2014*; *Parkinson, Coffroth & LaJeunesse, 2015*). Therefore, clustering all sequences together may underestimate diversity by collapsing different species into the same OTU. Conversely, treating each ITS2 sequence as a unique *Symbiodinium* type may overestimate diversity due to intragenomic variation (*Thornhill, LaJeunesse & Santos, 2007*). Within-sample clustering may better approximate true diversity by exploiting key assumptions regarding the distribution of symbionts among samples, and sequences among symbionts. These assumptions include: (1) that most coral colonies are dominated by a single *Symbiodinium* type (*LaJeunesse & Thornhill, 2011*; *Pettay et al., 2011*; *Baums, Devlin-Durante & LaJeunesse, 2014*; but see *Silverstein, Correa & Baker, 2012*), and (2) that different numerically dominant ITS2 sequence variants identify different *Symbiodinium* types, even when they differ by only one nucleotide (*Sampayo, Dove & LaJeunesse, 2009*). These assumptions suggest that when closely related ITS2 sequences (i.e., that are >97% similar; *Arif et al., 2014*) occur within the same sample, they are more likely to be intragenomic variants, but when they are numerically dominant in different samples, they are more likely to represent distinct *Symbiodinium* taxa. Accordingly, clustering sequences at 97% similarity within each sample independently may collapse variability that is more likely to be intragenomic (i.e., occurring within a sample), while maintaining variability that is more likely to be interspecific (i.e., occurring in different samples).

We demonstrate this within-sample clustering approach with an analysis of *Symbiodinium* communities in ten host species across the north and south shores of St. John. We explore ecological patterns that can be revealed by large-scale metabarcoding datasets, including (1) testing for whole-community differences associated with north and south shore locations, (2) analyzing coral-*Symbiodinium* metacommunity association networks, and (3) quantifying the variability in symbiont communities (i.e., beta diversity, or 'symbiosis flexibility') among individuals within a host species. The large quantity of data, and the kinds of analytical approaches facilitated by ITS2 metabarcoding, has the potential to revolutionize understanding of *Symbiodinium* metacommunity ecology in reef corals and other taxa harboring similar symbionts. More generally, the present study demonstrates how ecological knowledge can inform bioinformatic analyses using marker genes for which sequence diversity does not map directly to species.

# MATERIALS AND METHODS

## Sample collection and environmental conditions

Eighty-four tissue samples were collected from ten species of scleractinian and milleporine corals at seven sites around St. John, US Virgin Islands, USA (Fig. 1), between August 7th and 9th, 2012, as permitted by the Virgin Islands National Park (permit VIIS-2012-SCI-0017). Hosts were sampled at sites on both the north and south shores of St. John (except
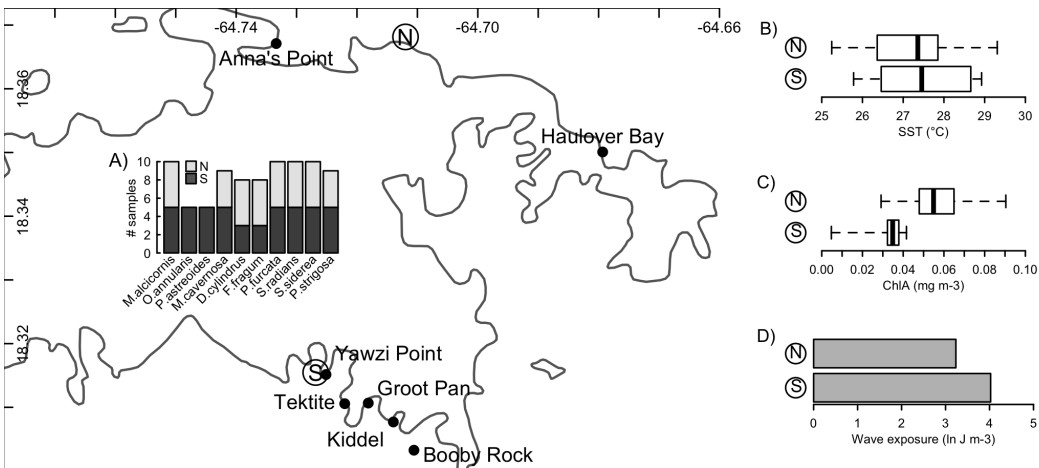

**Figure 1** **Sampling locations and environmental conditions.** Sampling of *Symbiodinium* communities from ten host species (A) on the north and south shores of St. John, US Virgin Islands, and the broad-scale physical conditions associated with these two environments (B–D). Mapped points along each shore indicate locations from which biological samples were collected, while 'N' and 'S' indicate locations from which environmental data representative of the north and south shores were obtained. Insets present box-plots of sea-surface temperatures (B) and chlorophyll a concentrations (C), and a barplot of the sum to-tal wave exposure (D) recorded at these two locations. Coastline data was obtained from NOAA GSHHG database.

for *O. annularis* and *P. astreoides* that were only sampled on the south shore). Samples were collected on snorkel from 3–7m depth, and colonies to be sampled were selected haphazardly in upward-facing locations exposed to ambient irradiance. Tissue biopsies (∼5-mm diameter and ∼5-mm depth) were removed from coral colonies with metal borers, and the holes were plugged with non-toxic modeling clay.

Broad-scale differences in the hydrographic regimes at points representative of the north and south shores of St. John (North: 18.368141°N, 64.711967°W; South: 18.315465°N, 64.726840°W) were evaluated from remote sensing tools to determine sea surface temperature (°C), surface chlorophyll concentrations (mg chl a m$^{-3}$), and integrated wave exposure on a power scale (J m$^{-3}$). These features were selected as they corresponded to anecdotal observations of differences between the water masses on both shores, conditions known to affect coral performance, *Symbiodinium* biology (i.e., SST (*Coles & Jokiel, 1977*) wave regime (*Atkinson, Kotler & Newton, 1994*)), and the supply of food resources that play important roles in coral nutrition (*Houlbrèque & Ferrier-Pagès, 2009*). SST and chlorophyll were determined using data from the MODIS sensor on the Aqua satellite. SST was evaluated from nighttime records, and chlorophyll from ocean colour, both of which were obtained at 1 km spatial resolution for each month, averaged over the years 2003–2010, from the IMaRS website (http://imars.usf.edu/modis); if data were not available at the chosen coordinates (above), values from the nearest available pixel were used. Boxplots were generated from the 12 monthly climatological values obtained for each response (Fig. 1).

Wind-driven wave exposure on a power scale for a given site is dependent on the wind patterns and the configuration of the coastline, which defines the fetch. To calculate wave

exposure, wind speed and direction at each location were acquired from the QuickSCAT (NASA) satellite scatterometer from 1999 to 2008 at 25 km spatial resolution (*Hoffman & Leidner, 2005*). Coastline data were obtained from the Global Self-consistent, Hierarchical, High-resolution, Shoreline (GSHHS v 2.2) database, which provides global coastline at 1:250,000 scale (*Wessel & Smith, 1996*). From these data, wave exposure was calculated using the methods based on wave theory (after *Chollett & Mumby, 2012*) for 32 fetch directions (equally distributed through 360°). Total wave exposure (summed over all directions) was calculated in R using the packages maptools (*Bivand & Lewin-Koh, 2016*), raster (*Hijmans, 2016*), rgeos (*Bivand & Rundel, 2016*), and sp (*Bivand, Pebesma & Gomez-Rubio, 2013*).

## ITS2 sequencing and bioinformatics

Coral tissue samples were preserved in ∼500 μL Guanidinium buffer (50% w/v guanidinium isothiocyanate; 50 mM Tris pH 7.6; 10 μM EDTA; 4.2% w/v sarkosyl; 2.1% v/v $\beta$-mercaptoethanol) and shipped to the Hawaii Institute of Marine Biology (HIMB). Genomic DNA was extracted from each coral tissue sample following a guanidinium-based extraction protocol (*Cunning et al., 2015*), and sent to Research and Testing Laboratory (Lubbock, TX) for sequencing of ITS2 amplicons ('itsD' and 'its2rev2' primers from *Stat et al., 2009*) on the Illumina MiSeq platform with $2 \times 300$ paired-end read chemistry.

Paired reads from each sample (provided in .fastq format by Research and Testing Laboratory) were merged using illumina-utils software (*Eren et al., 2013b*) with an enforced Q30-check and an overlap $\geq 150$ bp with $\leq 3$ mismatches required to retain a sequence. Chimeric sequences were removed using usearch61 (*Edgar, 2010*) implemented in QIIME (*Caporaso et al., 2010*). Primers were trimmed using cutadapt (*Martin, 2011*) allowing three mismatches, and only sequences with both forward and reverse primer matches and length $\geq 250$ bp after trimming were retained. Subsequently, three different clustering approaches, each based on the uclust algorithm (*Edgar, 2010*) and implemented in QIIME, were used to group ITS2 sequences into operational taxonomic units (OTUs): (1) clustering at 100% identity, (2) clustering at 97% identity across samples (i.e., sequences from all samples clustered together), and (3) clustering at 97% identity within samples (i.e., sequences from each sample clustered independently). For each 97% cluster, the most abundant sequence variant was chosen as the representative sequence, and within-sample clusters were merged across samples if their representative sequences were 100% identical. After removing singleton clusters, representative sequences for each OTU were assigned taxonomy by searching a custom reference database of *Symbiodinium* ITS2 sequences using the Needleman-Wunsch global alignment algorithm implemented in the Biostrings package (*Pagès et al., 2016*) in R (*R Core Team, 2014*). Each OTU was then assigned a name corresponding to the reference sequence(s) with the highest alignment score; if the match was <100%, the assignment was given a unique superscript. If the match was <90%, the sequence was blasted to the NCBI nt database and omitted if the top hit did not contain "*Symbiodinium*".

The reference database comprised *Symbiodinium* ITS2 sequences downloaded directly from NCBI. These sequences included those used in previous reference databases (*Cunning*

*et al., 2015*) supplemented with additional sequences of *Symbiodinium* from Caribbean corals (*Finney et al., 2010*; *Green et al., 2014*). Reference sequences were separated and aligned by clade using muscle (*Edgar, 2004*), and trimmed to equal length using the o-smart-trim command from oligotyping software (*Eren et al., 2013a*), before being reconcatenated into a single fasta file. The bioinformatic pipeline used here was implemented using a series of bash scripts, which can be found in the data archive accompanying this paper (*Cunning, 2017*). These scripts are provided along with all raw data and a Makefile, which can be executed to fully reproduce the present analysis.

### *Symbiodinium* data analysis

OTU tables, sample metadata, and taxonomic data were imported into R using the phyloseq package (*McMurdie & Holmes, 2013*) to facilitate downstream analyses. OTU tables were filtered to remove any OTU that was not observed at least 10 times, or any sample with <200 sequences (to remove possible sequencing errors and samples that did not sequence well), and counts were transformed to relative abundance. Permutational analysis of variance (PERMANOVA) was used to test for differences in *Symbiodinium* community composition between the north and south shores of St. John within each coral species.

Network analysis and visualization was performed in R using the igraph package (*Csárdi & Nepusz, 2006*). Networks were created featuring 'dominant' (>50% relative abundance) and 'abundant' (>1% relative abundance) OTUs, with weighted edges proportional to the number of individuals within a species in which a symbiont OTU occurred. A 'background' symbionts network was also created with unweighted edges defined based on the presence of a symbiont at <1% relative abundance in at least one individual within the species. To simplify visualization of the background symbiont network, clade D OTUs were merged into a single node under the assumption that clade D in the Caribbean comprises a single species (*Symbiodinium trenchii*; *LaJeunesse et al., 2014*), and symbiont nodes connected to ≤2 coral species were removed from the network. All network layouts were constructed based on the Fruchterman–Reingold algorithm (*Fruchterman & Reingold, 1991*).

Beta diversity (sensu *Anderson, Ellingsen & McArdle, 2006*) was calculated as the multivariate dispersion of samples within a coral species. Principal coordinate analysis of Bray–Curtis dissimilarities of square-root transformed OTU counts was used to calculate average distance-to-centroid values for each species, which were then compared statistically by a permutation test. This analysis was implemented using the betadisper function in the vegan package (*Oksanen et al., 2016*), based on *Anderson, Ellingsen & McArdle(2006)*.

## RESULTS

### Comparison of clustering approaches

The 84 coral samples of ten host species generated 1,490,813 sequences after merging paired reads, removing chimeric sequences, and trimming primers. After clustering, OTUs with <10 sequences and samples with <200 sequences were excluded, leaving 80 samples for downstream analysis. The number of OTUs resolved, as well as the number of sequences per OTU and per sample, depended on the clustering approach (Table 1). More OTUs were resolved as the clustering resolution increased (i.e., 97% OTUs across samples <97%

**Table 1  Summary statistics for each clustering approach.**

|  | 97% OTUs across samples | 97% OTUs within samples | 100% OTUs |
|---|---|---|---|
| Number of OTUs | 94 | 106 | 4,718 |
| Range of OTU counts | $10-74,2671$ | $10-47,2752$ | $10-17,1212$ |
| Range of reads per sample | $706-16,9884$ | $707-16,9890$ | $485-97,003$ |
| Geometric mean (*/ GSD) reads per sample | 13,141 */ 2 | 13,137 */ 2 | 8,141 */ 2 |

OTUs within samples <100% OTUs), with fewer reads per OTU (Table 1). The number of sequences per sample was less for the 100% OTU approach because more OTUs were filtered out of the dataset by the minimum threshold count of 10 reads per OTU.

A subset of samples was selected for comparative analysis of the community structure resolved by the three clustering approaches (Fig. 2). Despite being dominated by different sequence variants (Fig. 2A), across-sample clustering assigned the same dominant OTU to many of these samples (Fig. 2B), whereas within-sample clustering assigned them different OTUs (Fig. 2C) corresponding to their dominant sequence variants. Overall, across-sample clustering collapsed more sequence diversity into a single OTUs, while within-sample clustering resolved more distinct OTUs that associated with particular host species. Because the outcomes of the latter approach are more consistent with the current understanding of ITS2 sequence diversity as it links to *Symbiodinium* biology and ecology (see Discussion), the remainder of the results is presented using the 97% within-sample clustering approach. Detailed analysis and visualizations comparing the outcomes of the three clustering approaches for all individuals of each coral species are available in *Cunning (2017)*.

### *Symbiodinium* community composition

Within the set of coral samples, clade B had the highest relative abundance (46.6%), followed by clade C (41.3%), A (10.5%), D (1.5%), and G (0.1%). The number of OTUs within each clade followed a similar pattern, with the highest number in clade B (66), followed by clade C (25), A (6), D (5), and G (4). The distribution of these clades within each sample is shown in Fig. 3.

### Differences in *Symbiodinium* between shores

The environmental conditions broadly characterizing the north and south shores are presented in Fig. 1. From 2003 to 2010, the north shore of St. John was characterized by slightly lower sea surface temperatures, higher chlorophyll a concentrations, and lower wave exposure, relative to the south shore. No significant differences in *Symbiodinium* community composition between the north and south shores were detected for any host species (Table 2). However, qualitative differences were apparent: *Porites furcata* was dominated by either clade A or clade C *Symbiodinium* on the south shore, but only by clade C on the north shore (difference between shores PERMANOVA: $p = 0.056$). *Siderastrea siderea* was dominated by either clade C or clade D on the south shore, but only by clade C on the north shore (difference between shores PERMANOVA: $p = 0.071$).

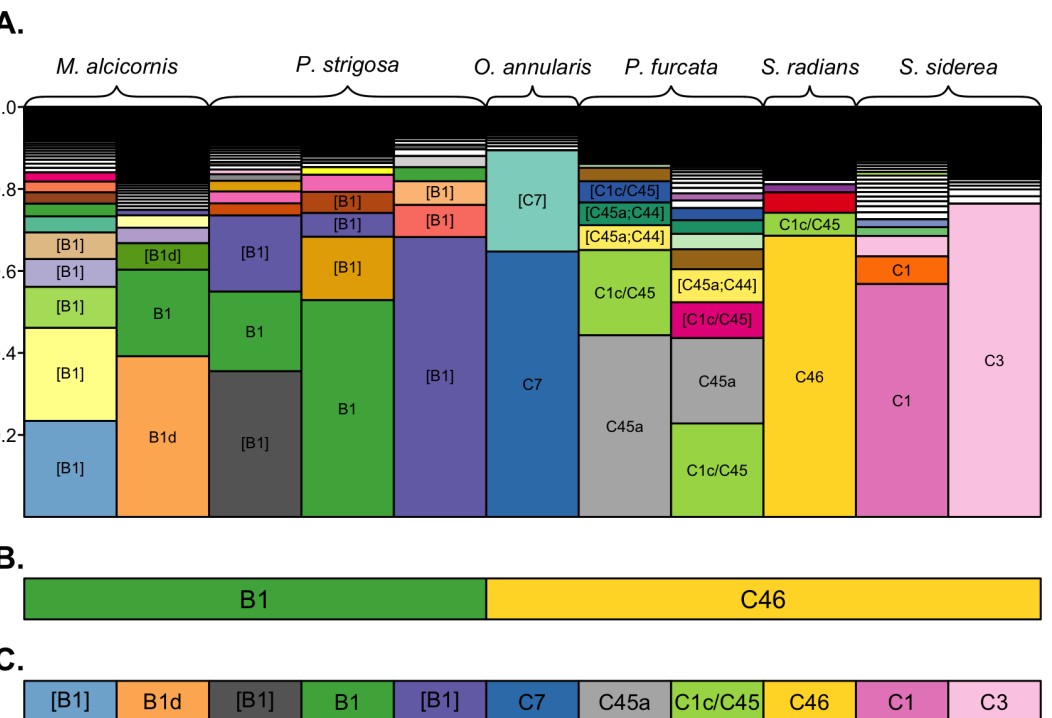

**Figure 2 OTU assignments by different clustering approaches.** Comparison of dominant OTUs assigned by 100%, 97% across-sample, and 97% within-sample clustering approaches. (A) The composition of unique sequence variants (i.e., 100% OTUs) in each sample, sorted by relative abundance, and filled by unique colors corresponding to unique sequence variants. Sequence variants are annotated with the name of their best-matching reference sequence, with brackets indicating the match is not exact (only OTUs with a relative abundance >0.05 are annotated for clarity). The dominant OTUs within each sample assigned by (B) 97% across-sample clustering and (C) 97% within-sample clustering are shown by rectangles below each bar, with fill colors matching the unique sequence variants presented in (A) to indicate the representative sequence of the assigned OTU. Colored rectangles that span multiple bars indicate that the corresponding samples were assigned the same dominant OTU.

## Network analysis of *Symbiodinium* metacommunity

Patterns of association between hosts and *Symbiodinium* were analyzed using networks focusing on "abundant" (>1% relative abundance; Fig. 4), "dominant" (>50% relative abundance; Fig. 5A), and "background" (<1% relative abundance; Fig. 5B) symbionts in each host species, based on 97% within-sample clustering. (Networks for individuals within each species are available in *Cunning (2017)*. In the network for abundant symbionts (Fig. 4), 37 OTUs were observed. *P. strigosa* associated with the greatest number of OTUs ($n = 17$), followed by *F. fragum* ($n = 5$), *M. alcicornis* ($n = 5$), and *D. cylindrus* ($n = 4$); these primarily comprised OTUs closely related to B1 and B19. In contrast, only a single OTU occurred at >1% relative abundance in both *P. astreoides* (A4) and *M. cavernosa* (C3). All coral species hosted at least one "abundant" *Symbiodinium* OTU that was also abundant in at least one other coral species, except for *S. radians*. This species was dominated by C46, but also contained *Symbiodinium* closely related to B1 and B19, as well as B5.

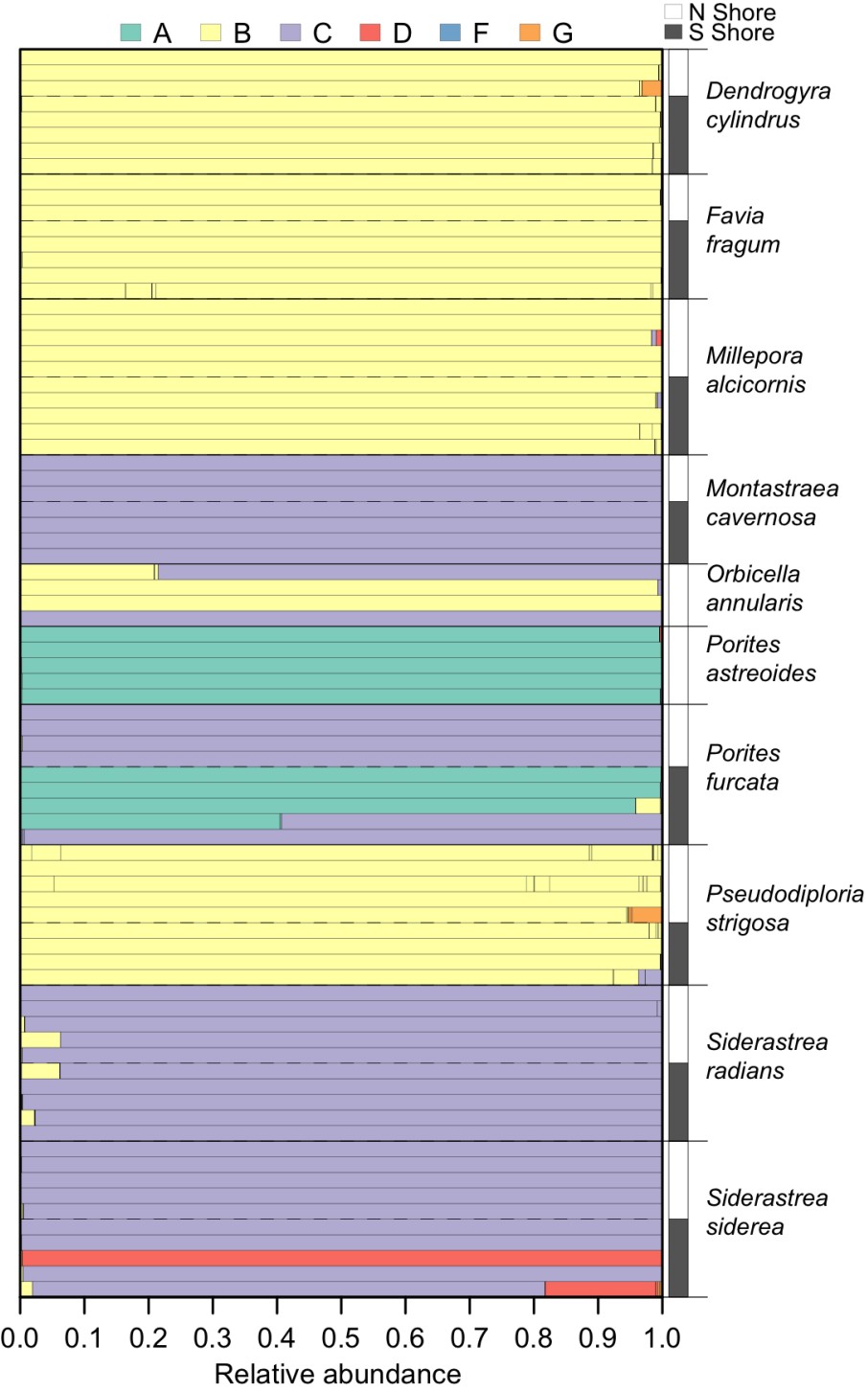

**Figure 3 *Symbiodinium* community composition of each sample.** Samples are plotted as horizontal bars, sorted by species and shore of collection (north vs. south). Individual segments of each bar represent 97% within-sample OTUs, colored by *Symbiodinium* clade identity.

**Table 2  Differences in *Symbiodinium* communities within and between shores.** Mean overall, within-shore ('within'), and between-shore ('between') Bray–Curtis dissimilarities of the *Symbiodinium* communities in each host species, and PERMANOVA tests (partial $R^2$ and $p$-values) for a difference between shores. Between-shore tests were not possible for *O. annularis* and *P. astreoides* since they were only sampled from one shore.

| | n | Overall | Within | Between | $R^2$ | $p$ |
|---|---|---|---|---|---|---|
| *Millepora alcicornis* | 10 | 0.604 | 0.653 | 0.564 | 0.037 | 0.809 |
| *Orbicella annularis* | 4 | 0.632 | 0.632 | – | – | – |
| *Porites astreoides* | 5 | 0.004 | 0.004 | – | – | – |
| *Montastraea cavernosa* | 7 | 0 | 0.001 | 0 | 0.118 | 1 |
| *Dendrogyra cylindrus* | 8 | 0.017 | 0.016 | 0.019 | 0.188 | 0.268 |
| *Favia fragum* | 8 | 0.251 | 0.309 | 0.202 | 0.086 | 0.75 |
| *Porites furcata* | 9 | 0.703 | 0.555 | 0.821 | 0.345 | 0.056 |
| *Siderastrea radians* | 10 | 0.03 | 0.031 | 0.029 | 0.064 | 0.953 |
| *Siderastrea siderea* | 10 | 0.512 | 0.541 | 0.489 | 0.061 | 0.467 |
| *Pseudodiploria strigosa* | 9 | 0.789 | 0.71 | 0.851 | 0.242 | 0.079 |

In the network for dominant symbionts, 15 different *Symbiodinium* OTUs were included, defined as those occurring at >50% relative abundance within a host (Fig. 5A). A single OTU identical to *Symbiodinium* B1 was the most prevalent dominant symbiont, dominating all *D. cylindrus* samples and many *P. strigosa* (4 of 9), *O. annularis* (2 of 4), *M. alcicornis* (6 of 10), and *F. fragum* (8 of 9) samples. Other closely-related clade B OTUs (similar to B1, B1d, and B19) occasionally dominated *P. strigosa*, *M. alcicornis*, and *F. fragum*, while *O. annularis* was dominated just as often by *Symbiodinium* C7. After B1, the next most prevalent dominant symbionts were *Symbiodinium* C3, which dominated all *M. cavernosa* and most *S. siderea* (7 of 10), and *Symbiodinium* A4, which dominated all *P. astreoides* and 3 of 10 *P. furcata*. *P. furcata* and *S. siderea* were each occasionally dominated by other symbiont OTUs, including C45 or C45a (*P. furcata*), and C1 or D1 (*S. siderea*). All coral species were dominated by at least one *Symbiodinium* OTU that also dominated at least one other coral species, except for *S. radians*, which was exclusively dominated by C46.

There were six symbionts that occupied a 'background' niche, defined as <1% relative abundance in three or more coral species (Fig. 5B). A member of *Symbiodinium* clade D (which has been described as *S. trenchii* (LaJeunesse et al., 2014)) was found at background levels in the greatest number of host species (7), followed by A4 and C3 (both in 6 taxa). C31, C46, and B1 were each detected at background levels in four host species.

## Symbiotic flexibility of host species

Symbiont communities from each individual host were ordinated in multivariate space, and the dispersion of individuals within each host species (i.e., average distance to centroid) was calculated as a metric of symbiosis 'flexibility'. Higher values indicate hosts with greater variability in symbiont community composition among individuals, and lower values indicating hosts with greater uniformity. This metric does not reflect how many *Symbiodinium* OTUs are hosted by a given coral species, but rather how much variation in *Symbiodinium* community composition occurs among different individuals. This metric

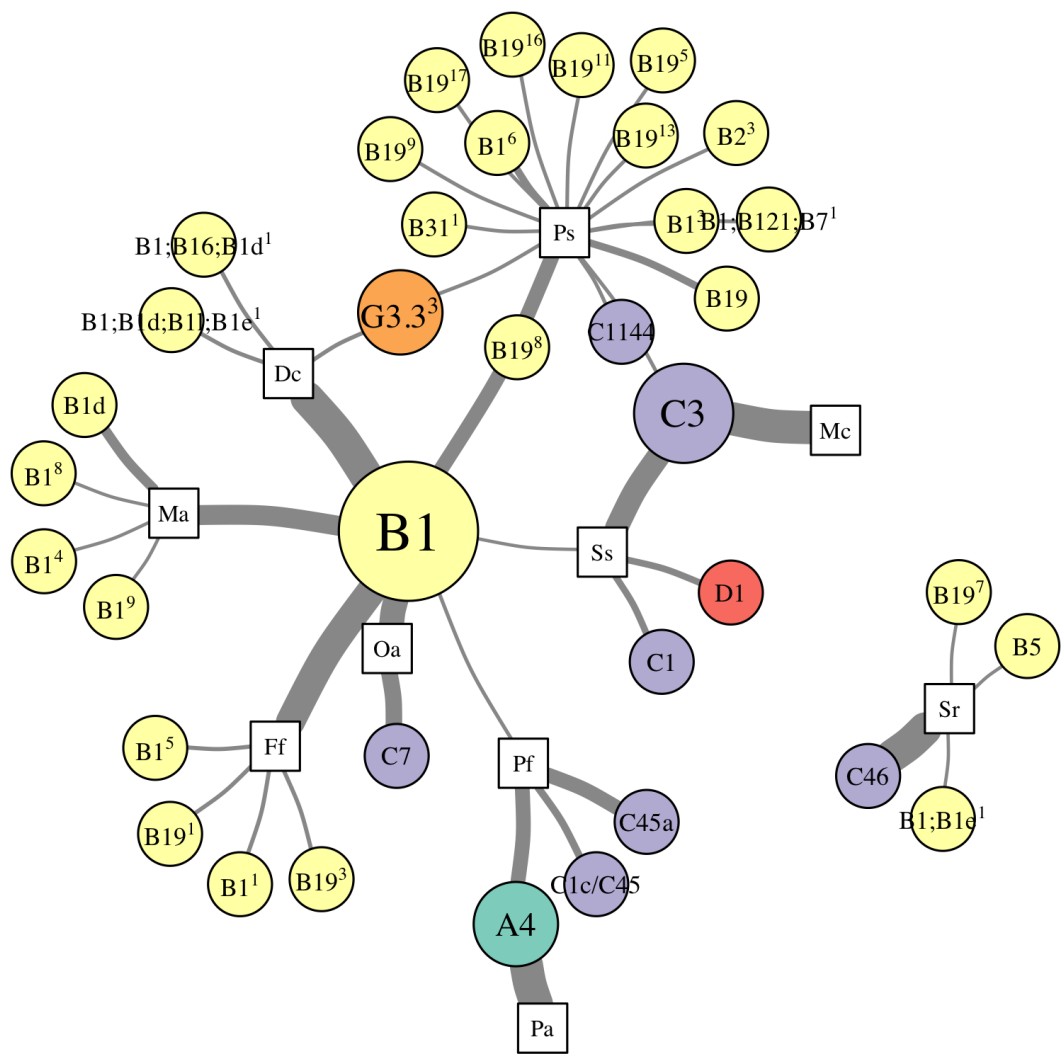

**Figure 4** **Network analysis of abundant *Symbiodinium* taxa in each coral species.** *Symbiodinium* OTUs (circular nodes) are connected to each coral species (square nodes) in which they ever occurred at >1% relative abundance within an individual; thickness of edges (i.e., links between coral and symbiont nodes) is relative to the proportion of individuals within the coral species in which the *Symbiodinium* OTU was present at >1%. Symbiont nodes are colored by clade identity, and sized proportionally to the number of coral species in which they were found.

revealed significant differences in symbiotic flexibility among host species (Fig. 6). The highest distance to centroid was found in *P. strigosa*, *P. furcata*, *M. alcicornis*, *S. siderea*, and *M. annularis*, and the lowest was found in *M. cavernosa*. Flexibility was also low in *D. cylindrus*, *S. radians*, and *P. astreoides*, and intermediate in *F. fragum*.

## DISCUSSION

### Bioinformatic analysis of ITS2 diversity

The structure of *Symbiodinium* communities that was detected in host species was impacted by the bioinformatic approach used to analyze ITS2 sequence data. Comparing taxonomic

A.)  B.)

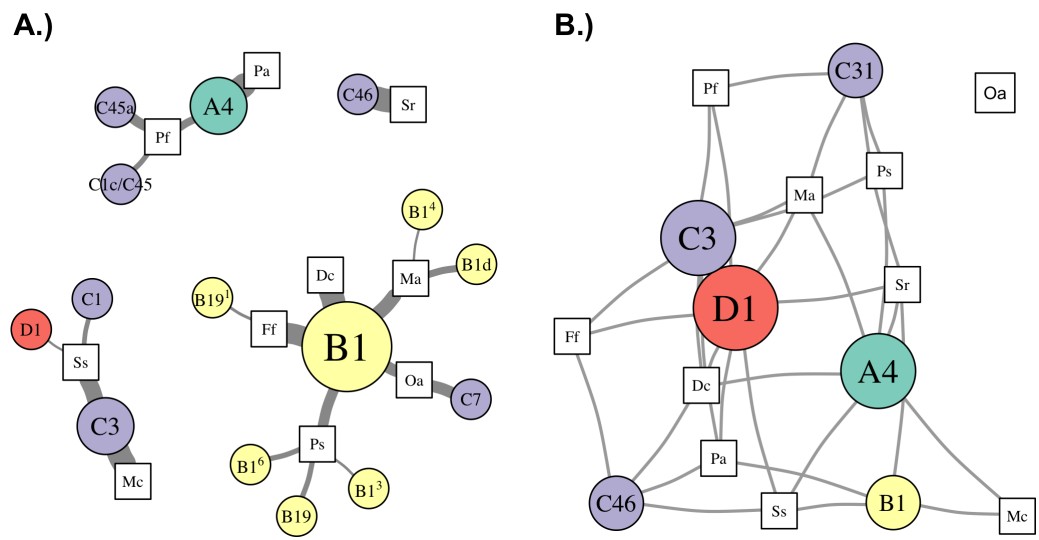

**Figure 5  Network analysis of dominant and background symbionts.** Network analysis of dominant (A) and background (B) *Symbiodinium* taxa in each coral species (see Materials and Methods for details on network construction). Symbiont nodes are colored by clade identity, sized proportionally to the number of coral species to which they are connected.

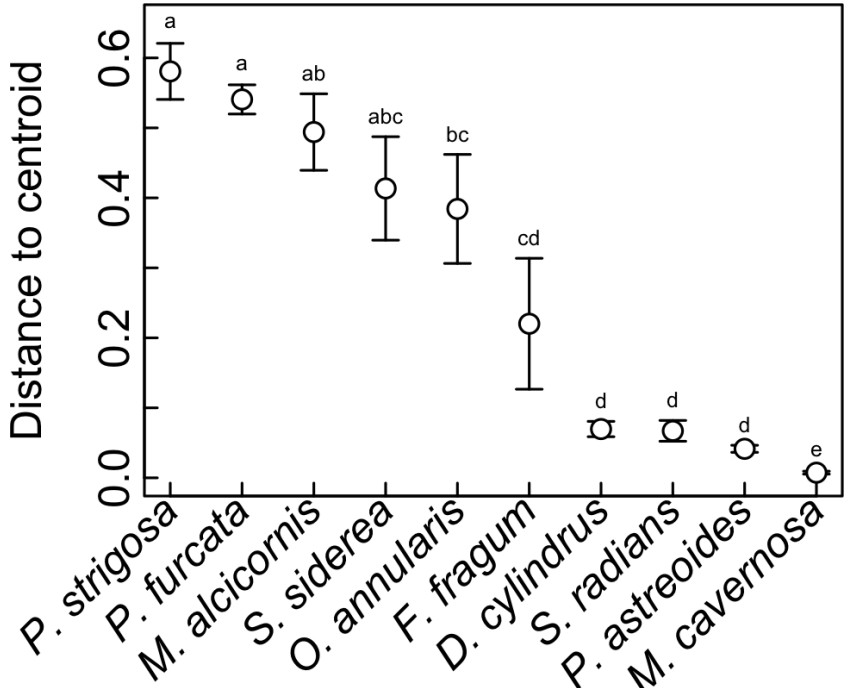

**Figure 6  Symbiotic flexibility (beta diversity) of host species.** Symbiotic flexibility in coral species quantified as multivariate dispersion of *Symbiodinium* community composition (mean distance to centroid) ±SE. Host taxa that do not share a letter are significantly different ($p < 0.05$).

assignments of the OTUs generated by these approaches demonstrates that across-sample clustering often assigns the same dominant OTU to samples with different dominant sequence variants (Fig. 2), while within-sample clustering assigns them distinct OTUs that reflect the dominant variant in the sample. This occurs because OTU identity is determined by the most abundant member sequence, and when members occur across many samples (i.e., during across-sample clustering), each member's abundance reflects the number (and sequencing depth) of samples in the dataset that contain it. Therefore, a particular sequence variant from a more extensively-sampled and/or deeply-sequenced host may determine the identity of an OTU that includes closely-related but different sequences in other samples, even though they do not actually contain the given sequence. Such merging of OTUs across samples is undesirable, since even single nucleotide differences in a sample's dominant ITS2 variant may indicate different *Symbiodinium* species (*Sampayo, Dove & LaJeunesse, 2009*).

Therefore, even though clustering at 97% similarity may be needed to collapse intragenomic variation within a single *Symbiodinium* (*Arif et al., 2014*), when such clustering is applied across a dataset potentially comprised of many closely related *Symbiodinium*, it is likely to also collapse interspecific variation and underestimate diversity. This limitation was also encountered by *Arif et al. (2014)* when clustering closely-related clade C sequences across samples from multiple hosts. Despite being comprised mostly of C41 variants, ∼3,000 ITS2 sequences from *Acropora hemprichii* were subsumed into a single OTU that was named C1 after the dominant variant among the ∼7,000 sequences from *Pocillopora verrucosa* with which they were clustered. In this case (and as shown in Fig. 2), clustering sequences across samples leads to the conclusion that samples with different dominant sequence variants are dominated by the same *Symbiodinium* OTU. This outcome occurs because the identity of OTUs in any given sample may be determined by other samples in the dataset; indeed, using this approach, the same sample and sequence assemblage may receive different OTU assignments if it were part of a different set of samples.

Meanwhile, clustering sequences within each sample independently assigns OTUs that reflect only the assemblage of sequences within that sample, and therefore does not depend on the presence or abundance of sequences from other samples included in the analysis. The outcome of within-sample clustering of ITS2 sequences as applied herein better reflects patterns of *Symbiodinium* diversity and ecology that have been revealed by more variable markers (psbA$^{ncr}$) and microsatellites (*Finney et al., 2010*; *Thornhill et al., 2014*); namely, that clades B and C in the Caribbean comprise numerous different *Symbiodinium* species that tend to associate with different coral host taxa. Indeed, only within-sample clustering assigned different dominant clade C OTUs to samples of *O. annularis* and *S. siderea* (Fig. 2), which have been differentiated based on other markers (*Thornhill et al., 2014*). The within-sample clustering approach should also reflect patterns that would be observed by sequencing dominant ITS2 bands from denaturing gradient gel electrophoresis (DGGE; the most commonly use method of describing *Symbioinium* diversity prior to metabarcoding), since both methods rely on numerically dominant sequence variants to assign taxonomy. Furthermore, metabarcoding overcomes the primary limitations of the DGGE method by providing more quantitative data and sensitive detection of low abundance taxa

(*Quigley et al., 2014*). Therefore, we recommend a within-sample clustering approach for metabarcoding studies where many different *Symbiodinium* types are likely to be encountered, but suggest that additional work with *Symbiodinium* cultures would help further validate this method.

Importantly, the assumptions underlying a within-sample clustering approach—that samples typically contain one *Symbiodinium* species that can be diagnosed by its most abundant intragenomic ITS2 variant—will not always be met. Indeed, multiple *Symbiodinium* frequently co-occur in single coral colonies (*Silverstein, Correa & Baker, 2012*), which undermines support for assuming that variation within a sample is intragenomic. However, co-occurring *Symbiodinium* in many cases are members of different clades, whose ITS2 sequences are divergent enough to be resolved separately by 97% clustering. Thus, only when very closely related types co-occur in a sample (e.g., *Sampayo et al., 2007*; *Wham & LaJeunesse, 2016*) is this approach likely to fail. Other problematic cases may occur when multiple intragenomic sequence variants are co-dominant (i.e., comprise nearly equal proportions of the rDNA array), such that slight differences in their relative abundance may lead to different OTU assignments in samples with otherwise highly similar sequence assemblages. These challenges may be overcome by basing taxonomic assignment on more complex criteria than just the most abundant sequence variant, such as multiple co-dominant variants, or more evolutionarily derived variants. Alternatively, other markers (mitochondrial, microsatellites and flanking regions) may help overcome these taxonomic challenges (*Stat et al., 2012*).

While no treatment of ITS2 data can provide species-level descriptions of *Symbiodinium* communities (since the marker itself does not provide this resolution), within-sample clustering nevertheless accommodates known evolutionary complexities in the ITS2 locus to provide community descriptions that better reflect species-level diversity than either across-sample clustering or no clustering at all. When treated with an appropriate bioinformatic approach, ITS2 metabarcoding can provide a comprehensive and quantitative analysis of *Symbiodinium* ITS2 diversity, and can be applied across host species to rapidly survey the *Symbiodinium* metacommunity on a reef-wide scale. To complement this field-based study, we suggest that sequencing of prescribed mixtures of cultured *Symbiodinium* species could strengthen support for this approach, and identify cases in which it could be improved.

### *Symbiodinium* metacommunity ecology in St. John

In the ten host species sampled, the most prevalent *Symbiodinium* were members of clades B and C, which is consistent with previous analyses of *Symbiodinium* diversity on shallow reefs in this region (*LaJeunesse, 2005*; *Correa et al., 2009*; *Finney et al., 2010*; *Edmunds et al., 2014*). Less frequent associations between hosts and members of clades A and D were observed, although clade A dominated *P. astreoides* and some *P. furcata*, and clade D dominated one *S. siderea*. Finally, clade G was observed at low relative abundances (<5%) in some coral colonies, which likely would not have been detected without high-throughput sequencing.

While the environmental conditions differed between the north and south shores of St. John (e.g., wave exposure, temperature, and chlorophyll a), there were no statistically

significant differences in symbiont community composition in the corals sampled on either shore. This may partly be due to small sample sizes, since two species showed a trend for differences between shores: *S. siderea* and *P. furcata* were more frequently dominated by *Symbiodinium* in clades D and A (respectively) on the south shore, compared to the north shore (Fig. 2). *Symbiodinium* in clades D and A are typically associated with warm and variable temperatures (e.g., *Baker et al., 2004*; *Oliver & Palumbi, 2011*) and shallow habitats with high light intensities (*LaJeunesse, 2002*). Therefore, it is consistent with expectations that they were more prevalent on the south shore, where temperatures were slightly higher, and chlorophyll levels were lower, suggesting greater light penetration into the water column. However, the lack of differentiation between shores in the symbiont communities of most host species suggests that environmental differences at this relatively small scale are not major drivers of *Symbiodinium* community structure.

Network analysis of associations between *Symbiodinium* types and coral species revealed metacommunity-level patterns in *Symbiodinium* ecology. First, ITS2 type B1 was "abundant" (i.e., >1% relative abundance in a sample) in seven of ten host species (Fig. 4), and "dominant" (i.e., >50%) in five of ten (Fig. 5), suggesting it is a generalist symbiont in St. John. However, analysis of other markers such as chloroplast and microsatellite loci has revealed that *Symbiodinium* ITS2 type B1 in the Caribbean is comprised of multiple lineages that show high host species fidelity (*Santos et al., 2004*; *Finney et al., 2010*; *Parkinson, Coffroth & LaJeunesse, 2015*). Likewise, the apparent generalists *Symbiodinium* C3 (which dominated colonies of *S. siderea* and *M. cavernosa*) and *Symbiodinium* A4 (which dominated *P. astreoides* and *P. furcata*, Fig. 5) may also be comprised of multiple host-specialized lineages (*Thornhill et al., 2014*), which could be revealed using higher resolution genetic markers.

In addition to hosting apparent generalist ITS2 lineages, most coral species sampled in St. John (except *M. cavernosa* and *P. astreoides*) hosted abundant *Symbiodinium* OTUs that were not abundant in any other corals, and are thus apparently more host-specific. Unique OTUs in the B1-radiation were abundant in *M. alcicornis* and *D. cylindrus*, while others from both the B1- and B19-radiations (see *LaJeunesse, 2005*) were abundant in *F. fragum*, *S. radians*, and *P. strigosa*. Several clade C OTUs similarly were only abundant in one species: C1144 in *P. strigosa*, C7 in *O. annularis*, C1 in *S. siderea*, C1c/C45 and C45a in *P. furcata*, and C46 in *S. radians*. Interestingly, *S. radians* was the only species in which none of the abundant symbionts occurred in any other host, which may reflect the unique ecology of *S. radians* as the only study species that typically forms small, encrusting colonies as adults.

To reveal symbionts that consistently occupied a 'background' niche, we identified OTUs present at <1% relative abundance in samples of three or more host species. This distribution suggests that these symbionts can occupy a range of hosts, but are unlikely to be dominant symbionts, at least under prevailing environmental conditions. While it is possible that some of these may be free-living surface contaminants not in symbiosis with the host (*sensu* *Silverstein, Correa & Baker, 2012*), the most prevalent 'background' symbiont in St. John was a member of clade D (*Symbiodinium trenchii* (*LaJeunesse et al., 2014*)), which is known to proliferate within hosts during and after thermal stress

(*Thornhill et al., 2006*; *LaJeunesse et al., 2009*; *Silverstein, Cunning & Baker, 2015*), or in marginal environments (*LaJeunesse et al., 2010*). Other symbionts occupying a background niche in St. John, such as *Symbiodinium* A4 and C3, may similarly become dominant under different sets of environmental conditions. These background symbionts, therefore, may perform an important functional role within the metacommunity by broadening the fundamental niche that a host may occupy (*sensu Bruno, Stachowicz & Bertness, 2003*), thereby increasing its resilience to environmental change (*Correa & Baker, 2011*).

Indeed, the ability of corals to associate with different symbionts may be an important trait that mediates their sensitivity to stress (*Putnam et al., 2012*), and their ability to change symbionts over time in response to environmental change (*Baker, 2003*). While flexibility in symbiosis ecology has been quantified previously for individual corals (i.e., based on the diversity of symbionts co-occurring within a single host colony (*Putnam et al., 2012*)), here we quantify flexibility at the host species level (*Fabina et al., 2012*; *Fabina et al., 2013*) based on quantitative variability of symbiont community structure among multiple colonies (Fig. 6), a metric of beta diversity that can be statistically compared among host species (*Anderson, Ellingsen & McArdle, 2006*). This metric revealed that *M. cavernosa*, *P. astreoides*, *S. radians*, and *D. cylindrus* had the lowest symbiotic flexibility, meaning that all sampled individuals had similar symbiont community structure. In these host species, symbiont communities may be more constrained by host biology and/or less responsive to environmental variation. On the other hand, *P. strigosa*, *P. furcata*, *M. alcicornis*, and *S. siderea* had high flexibility, meaning that sampled individuals displayed more divergent symbiont communities. Community structure in these hosts may be more responsive to variability in the environment (e.g., *Kennedy et al., 2016*), or subject to greater stochasticity. While comparisons among studies are often confounded by methodology and location, the flexibility of these species in St. John is generally congruent with broad-scale patterns of symbiont associations recorded in the Coral Traits database (coraltraits.org; *Madin et al., 2016*). While future work should investigate whether the scope for symbiont community change over time within individuals is linked to variability among individuals, we suggest that the latter represents a useful metric of symbiosis flexibility that can be easily quantified using metabarcoding data.

## CONCLUSIONS

ITS2 metabarcoding and within-sample OTU clustering represents a powerful approach to quantitatively and comprehensively describe *Symbiodinium* metacommunity composition on coral reefs. The scale and resolution of datasets generated in this way facilitate new analytical applications that can address critical topics in coral symbiosis ecology, including changes in dominant and background symbionts across environmental gradients and over time, and the role of metacommunity processes in shaping *Symbiodinium* communities. Describing these trends has the potential to greatly advance understanding of coral responses to environmental change.

## ACKNOWLEDGEMENTS

Remote sensing observations of sea surface temperature and phytoplankton pigment concentration were processed at the Institute for Marine Remote Sensing at the College of Marine Science, University of South Florida, in St. Petersburg, Florida, and provided courtesy of Frank Müller-Karger and Iliana Chollett.

### Funding

Funding for this work was provided by the Long Term Research in Environmental Biology program of the US National Science Foundation (NSF) (DEB 13-50146). RC was supported by a NSF Postdoctoral Research Fellowship in Biology (NSF-PRFB 1400787). There was no additional external funding received for this study. The funders had no role in study design, data collection and analysis, decision to publish, or preparation of the manuscript.

### Grant Disclosures

The following grant information was disclosed by the authors:
Long Term Research in Environmental Biology program of the US National Science Foundation (NSF): DEB 13-50146.
NSF Postdoctoral Research Fellowship in Biology: NSF-PRFB 1400787.

### Competing Interests

The authors declare there are no competing interests.

### Author Contributions

- Ross Cunning conceived and designed the experiments, performed the experiments, analyzed the data, wrote the paper, prepared figures and/or tables.
- Ruth D. Gates conceived and designed the experiments, contributed reagents/materials/analysis tools, reviewed drafts of the paper.
- Peter J. Edmunds conceived and designed the experiments, performed the experiments, reviewed drafts of the paper.

### Field Study Permissions

The following information was supplied relating to field study approvals (i.e., approving body and any reference numbers):
Field collections were permitted by the Virgin Islands National Park (permit VIIS-2012-SCI-0017).

### Data Availability

Cunning R. (2017) Data for: using high-throughput sequencing of ITS2 to describe *Symbiodinium* metacommunities in St. John, U.S. Virgin Islands. Zenodo. 10.5281/zenodo.803992.

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
