# Peer review of "Using high-throughput sequencing of ITS2 to describe Symbiodinium metacommunities in St. John, US Virgin Islands"

_PeerJ, doi:10.7717/peerj.3472_

## Round 0.1 · original submission · Minor Revisions

All of the reviewers commend the work done in the paper and note that it will be significant for researchers in the field of Symbiodinium biology. The paper is very well written and provides a very useful contribution to the field. There are a number of minor revisions that are recommended by the different reviewers. Please consider these and address them in your resubmission.

·

Basic reporting

The paper is excellent in this regard. It is well written. The figures are informative and carefully presented. The literature is sufficiently cited and results are appropriately contextualized. The methods are transparent. The results and discussion are self contained. Results are relevant to the scope of the paper.

There are several references with missing details or inaccurate dates (e.g., LaJeunesse 2004 should be 2005; Thornhill et al. 2013 should be 2014; Correa et al. 2009 is missing volume and page information; same issue for Cunning et al. 2015a, Parkinson et al. 2015, Thornhill et al. 2014). This should be corrected.

Experimental design

The manuscript is excellent in this category as well. ITS2 data have been widely used to assess Symbiodinium diversity, however the marker is problematic for reasons outlined by Cunning, Gates, and Edmunds (insufficient resolution for species, multi copy, intragenomic variation, etc.). This has confounded the field for many years and led to a contentious debate about the nature of coral-Symbiodinium associations. The field has been moving away from earlier molecular approaches to next generation sequencing, yet the difficulties of working with the ITS2 remain. Several next gen sequencing studies adopted an OTU clustering approach based on sequence similarity, but this collapses and removes most of the ecologically and functionally meaningful data and is often little better than a clade-based analysis.

Here the authors present an innovative bioinformatic approach and apply it to 10 species of corals from St. John, US Virgin Islands. The key innovation is that clustering is done on a per colony basis and weighted towards the sequence that is numerically dominant within that colony prior to comparisons across samples or species. This reveals more ecologically meaningful patterns of diversity that are largely consistent with early ITS2 work as well as data from chloroplast genetic markers, while at the same time capturing patterns of diversity in the "cryptic" low-abundance OTUs. This is a considerable improvement over prior approaches. While I don't think the results produce a perfect separation of inter and intra genomic variability, I do think this is as close as the community will ever achieve using ITS2 in Symbiodinium. The only way I can imagine to improve upon the approach outlined here is to more widely sample the genomes of these symbionts.

The problem is well defined and important to the field. The work is preformed to a high technical standard. There are no ethical concerns; permit numbers are provided for the collection of corals for this study. The methods are detailed and transparent. The authors have provided the scripts and raw data via github to allow for the analysis to be repeated.

The only flaw I found is that the sample size of the 10 species of corals is relatively small (5 or fewer colonies per side of the island, with N and S sampling). I don't think this is a significant issue as the paper is presented more as a demonstration of the analytical approach than a comprehensive sampling of symbiont associations around St. John.

Validity of the findings

The paper is sound in this category. Considering the popularity of using ITS2 for diversity surveys, this paper should be a useful and highly-cited contribution. The results are robust, statistically sound, and consistent with prior knowledge about these associations. Conclusions are clearly stated. Assumptions and limitations are transparently acknowledged.

Beyond the comments above, I have several minor suggestions:

Paragraph ending on line 375: sampling with other markers (chloroplast, mitochondria, microsats and flanking regions, etc.) is another way to overcome the taxonomic challenges. I believe Stat et al. 2012 could be cited in support of this.

Line 381: As the authors noted earlier the ITS2 does not really give a comprehensive assessment of diversity. Suggest adding "ITS2" between "Symbiodinium" and "diversity" on line 382.

Line 382: I don't think it is fair to call this approach easy. It requires a great deal more technical expertise and resources than the older and more basic molecular approaches.

Additional comments

No additional comments.

Reviewer 2 ·

Basic reporting

There was sufficient field background and literature references, although I have made suggestions to some additional references that should be added (see general comments for author section).

Overall, the article structure is sound but some changes should be made concerning the presentation of Results and Discussion around the clustering approach. Specifically, the authors present very little results on the clustering approach (only lines 240-248), and instead use the Discussion to present these results. Starting lines 244: “Because of the outcomes of the latter approach are more consistent with the current understanding of ITS2 sequence diversity as it links to Symbiodinium biology and ecology (see Discussion).” This section of the manuscript reads as if the original version had Results/Discussion formatted together. Further clustering results need to be presented in the Results section. The Abstract presents this new clustering method as a main point of the manuscript (“Here, we demonstrate that generating operational taxonomic units by clustering ITS2 sequences at 97% similarity within, but not across, samples collapses sequence diversity that is more likely to be intragenomic, while preserving diversity that is more likely interspecific”) but really only dedicate 9 lines of Results to this. I would suggest 1) moving Results out of the Discussion and 2) adding more justification/reasoning/evidence to why this was chosen as the correct method and 3) adding an additional analysis simulating how co-occurrence of dominant symbionts would be effected by the within-clustering methods (as discussed in lines 361-363).

The manuscript is wholly self-contained, although suggestions have been made concerning the addition of results (see above).

Experimental design

Although I recommend publication, this manuscript suffers from seemingly trying to present two different stories, although neither is particularly strong. Firstly, this paper presents information on in-hospite Symbiodinium distributions across different environmental gradients, although no significant patterns emerge (which is fine as impact and novelty are not assessed and negative results are accepted in PeerJ). Secondly and most interestingly, it presents a new bioinformatic method to deal with intragenomic variation by clustering within-in samples at 97% identity. However, this study did not employ the optimal experimental design to test this method. Instead of using field data, the authors conclusions would have been more convincing if they had sequenced samples of known identity, i.e. a dilution series made up of cultured Symbiodinium and/or technical replicates to verify that comparisons across samples are still accurate even when each samples is clustered separately. Regardless, I still recommend publication as it does contribute to building the argument for this bioinformatic technique and presents new sequence data for Symbiodinium. However, because of the lack of some kind of standardized sequencing (dilution or technique replicates) the authors should clearly articulate in the discussion that further testing is needed to actually verify that this technique is the optimal method. The only reasoning really given to support their conclusions is that the resulting Symbiodinium diversity most closely resembles known diversity (lines 244-246: “Because the outcomes of the latter approach are more consistent with the current understanding of ITS2 sequence diversity as it links to Symbiodinium biology and ecology (see Discussion)”), which just really isn’t a rigorous enough approach.

Validity of the findings

No comment.

Additional comments

I recommend publication. This paper presents an interesting and novel bioinformatics method for addressing a common issue in coral reef/Symbiodinium biology. This study is important in that it provides evidence for clustering within samples at 97% as well as presenting compelling evidence (in conjunction with Arif et al. 2014) that clustering across samples at 97% does not artificially inflate estimates of Symbiodinium diversity. I enjoyed reading this paper as the writing was very clear and the figures were well thought-out and designed.

I commend the authors for the additional scripts and Github resources, which further added to the clarity of the manuscript.

I also applaud the authors for introducing a simple measure of Symbiodinium/host flexibility that has been generally lacking in this area. It will hopefully create a level of consistency across studies looking at the flexibility of Symbiodinium symbioses.

Some comments below that I hope will improve the manuscript:

Introduction:
line 38: Further studies should be cited here showing variation in communities over time, for example Thornhill et al. 2006a,b; Baker 2001; Baker 2004; Edmunds et al. 2014

line 36: add Bay et al. 2016

lines 40: Maybe add paper on adaptive bleaching hypothesis, Kinzie et al. 2001

lines 50: Recent taxonomic work with microsatellites should be added, for example, Parkinson et al. 2015 (Intraspecific diversity…) and Thornhill et al. 2014 (Host-specialist lineages…).

lines 53: Further work should be added here: Green et al. 2014, Kenkel et al. 2013, Quigley et al. 2016, 2017

lines 88-91: This line seems out of place in the Introduction. At this point (before Results are presented), this statement is speculation. If based on other work, this statement should be supported by other work or should be moved into the Discussion.

lines 90-91: *This statement is not necessarily true and should be modified or a reference should be added to validate this statement. Arif et al. 2014 show that clustering at 97% identity effectively collapses intragenomic variation, resulting in unique OTUs that represent biological entities and not intragenomic variants. The work from this manuscript actually shows that clustering at 97% identity across samples underestimates diversity! Therefore, the statement “Conversely, treating each ITS2 sequence as a unique Symbiodinium type overestimates diversity due to intragenomic variation” is misleading and does not seem to have any evidence behind it.
line 93: It should be clarified further that this assumption [1) that most coral colonies are dominated by a single Symbiodinium type (LaJeunesse and Thornhill, 2011; Pettay et al., 2011; Baums et al., 2014)] is controversial. For example, Silverstein 2012 thesis; Baker and Romanski 2007.
line 101: Goes back to the comment above regrading lines 90-91: Again, this statement needs some reference to back it up, otherwise it looks like speculation. Surely you can find papers employing similar clustering principles in the microbial literature to validate this statement.
General comment: The Introduction reads a bit long and speculative (probably due to the lack of some references as mentioned above). I would recommend condensing the last two paragraphs into one as they are repetitive.
For example, the authors start out lines 78-83 with stating the objectives:
“Here, we describe a Symbiodinium metacommunity associated with scleractinians and a Millepora hydrocoral in St. John, U.S. Virgin Islands, with the objectives of: 1) developing an appropriate… and 2) exploring network analysis…..
They then re-iterate this in the next paragraph, lines 105-109:
“Using within-sample clustering, we analyzed Symbiodinium communities in ten host species across the north and south shores of St. John. We explore ecological patterns that can be revealed by large-scale metabarcoding datasets, including 1) testing for whole-community differences associated with north and south shore locations, 2) analyzing coral-Symbiodinium metacommunity association networks…”
Materials and methods:

lines 198: what is the justification of the removing an OTU that did not appear at least 10 times? or 200 sequences? This justification and citations should be added.

line 205: Add citation as to why you decided to define “abundant” as >1%.
Results:

line 224: Do O. annularis and P. astreoides not occur on south side or just not sampled? I might have missed this in the methods, but just clarify why they are not found in Fig. 1

lines 240-248: Further results are needed here, including those that can be found in the Discussion. See comments from Basic Reporting and Validity of the findings sections. The main issue is that the only justification for using within-sample clustering is that is most closely resembles known diversity. If this is the main point of the manuscript (in abstract and the first topic addressed in the Discussion), more results and justification need to be presented. Fig.2 is a great figure, but the results could be presented more fully. For example, with the 100% clustering method, what are the other types reported (like the larger orange box above C1 in the first S. siderea sample). In the Discussion, you should also talk about what you think these other types represent in the 100% clustering. Are they all intragenomic variants? There is a lot of interesting information in Figure 2, but unfortunately it is not presented and not discussed much. An opportunity for another analysis is presented in lines 361-363 of the Discussion: an analysis showing what would happen if there was co-occurrence. Perhaps present a similar figure like Fig. 2 using samples like the two C types in sample 1 of Pseudodiploria strigosa from the north shore (figure 3).

lines 292: When blasted to the custom database or to NCBI, did the authors get D1 or other D type sequences? It would be interesting to see a bit more results concerning D sequences instead of just assuming for simplicity that all sequences must belong to D1a (as stated on line 210-212 due to LaJeunesse et al., 2014). This might be an opportunity to detect novel diversity.

line 303: Just from the figure 3, it looks like D. cylindrus would have high flexibility, but it was quantified as having low flexibility. Maybe a supplementary figure showing all the background types (and excluding the dominant type/types) would make this paragraph more convincing or easier to visualize? The number of types/nodes around Dc in Figure 4 also suggests a larger degree of flexibility.

Discussion:

Lines 311-326 reads as if it should be in the Results, especially given you are citing tables and figures in this section.

Lines 321-324 is not convincing….need some citation to support this statement.

Lines 345-348: essentially say that the within-sample method is correct because it aligns best with results of two studies: Finney et al. 2010 and Thornhill et al. 2013. This is not very rigorous. This would be a good area to clearly articulate in the discussion that further testing (dilution series/sequencing technical replicates) is needed to actually verify that this technique is the optimal method.

Line 349 - 351 was this in results? If not, it should be moved there.

lines 361-363: It would be nice to see an analysis showing what would happen if there was co-occurrence.

Section below line 384: I might have missed this but I generally felt that more information is needed comparing known information on the diversity of Symbiodinium and flexibility of associations with that of the species presented here (work by Fabina et al. 2012, 2013). You do this well in the paragraph 406- 416 when comparing B1 results to currently known information from Parkinson et al. 2015 and others. For example on line 269: P. strigosa is presented as the most flexible and D. cylindrus as the least flexible. How does this compare with information currently known on these species? Especially given that you are using a different clustering protocol, you should compare the diversity found here to other studies including: Orbicella annularis (Kennedy et al. 2015, 2016, Edmunds et al. 2014) and Orbicella faveolata, Orbicella franksi (Green et al. 2014) – similar species and methods

Lines 404-405: This sentence comes across as environmental differences are “not major drivers of Symbiodinium community structure,” which is not true in most cases (i.e. Davies et al. 2016 Ecological factors… amongst many others). It needs to be clearer that environmental differences are just not important here probably due to the relatively small magnitude of differences between north and south.

Line 409: suggesting it is a generalist symbiont at this location. That caveat should be added.

Line 443-444: Fabina et al. 2012 and 2013 should be discussed here.


References:
Formatting of species names needs to be checked as well as some minor edits to authors names, journals.

line 480: italic Symbiodinium
483: porites compressa italics
486: Symbiodinium italics
499: van Oppen, not Oppen MJH van
502: Nature Methods. m needs caps
511: italic Symbiodinium
521: Symbiodinium italics. not every word needs to be in caps here
529: italic Symbiodinium, orbicella annularis
535, 554, 575, 603, 621: not every word needs caps
539, 543, 546,555, 557, 563, 571, 580, 582, 594-596, 599, 601, 616, 625, 634, 636, 649, 653, 657 : italic Symbiodinium
547, 559: italic species names
610-611: caps and italics Symbiodinium
645: Thornhill all in caps. also remove n/as


Table 2: R2 and p should be R2 and p
Symbiodinium, O. annularis and P. astreoides in italics

·

Basic reporting

no comment

Experimental design

no comment

Validity of the findings

no comment

Additional comments

This paper makes a good attempt at addressing the long-standing problem of Symbiodinium diversity inference from ITS2 sequence variation. High-throughput sequencing offers a much faster and convenient means of assessing reef diversity than traditional electrophoretic and cloning methods; yet teasing out the true underlying diversity from the intra-genomic ‘noise’ remains a challenge when using multi-copy markers such as the ITS2. This paper proposes a possible solution involving clustering OTUs at the level of the individual coral colony.

The authors did a good job explaining the motivation for clustering OTUs within samples, since the alternative method of clustering across the entire set of samples has the undesired result of collapsing multiple taxa into single OTUs. However I do suspect that this has had the opposite effect of over-inflating the true taxonomic diversity a little. This is because, as the authors point out, intragenomic ITS2 variants are often co-dominant in Symbiodinium. In many examples this involves a commonly observed sequence such as C1 or C3 in coexistence with one or more similar taxon-specific, diagnostic ITS2 types. In such situations the proposed method would have a higher probability of erroneously generating two ore more OTUs where only one taxon exists. One solution that the authors may wish to consider (though I understand that it may not be logistically feasible) is to carry out additional direct sequencing of an alternative high-resolution marker such as the non-coding region of the psbA minicircle. Though this would unlikely help with the ‘background’ taxa such as the clade G type found here, it may help to validate the diversity estimate for the dominant taxa. While the high incidence of intragenomic variation/codominance is clearly an ongoing issue when using this marker, the authors clearly explained this limitation (lines 369-375) and I agree with their conclusion that more complex models that also take relative abundances of rare sequence variants into account are needed.

Overall I found the paper very well written, clear, relevant, and of interest to a wide audience. Despite its limitations I believe that this study is a step in the right direction, and provides a timely contribution to the field.

Some additional minor edits are suggested below

Lines 223-224. This sentence is a reiteration of the methods. Consider removing or integrating with the following sentence in ‘results style’. Also the sentence on lines 228-231 seems a little out of place here, optionally consider moving to the paragraph starting on line 255.

Lines 436-439 I would also offer an alternative explanation here, that the background symbionts are simply those that are living in high abundance in the water column or represent surface contamination. The HTS approach often picks up very low abundance sequences (hence the success of eDNA barcoding), so it seems possible that some of these low abundance types may not be symbiotic with the host.

Lines 296-300. This is a long sentence with a lot to take in. Perhaps consider breaking it in two.

Figure 1. The site labels on the figure seem quite cluttered and may need some rearranging. Also the wave exposure boxplot is different from the others – is it in fact a box plot or just a horizontal bar graph? Error bars would be useful (assuming the errors aren’t just so small I can’t seem them..)

Figure 5A seems like a repetition of Figure 4 and might be surplus to requirements. Figs 4 and 5 could optionally be merged.

---

## Round 0.2 · accepted · Accept

Thank you for your attention to the recommendations by the reviewers. This is an excellent manuscript and will be of great interest and use to the Symbiodinium community.

Thanks for choosing to publish in PeerJ!